# Comparison of Precipitation Parameterizations in Regional Climate Model (RegCM5): A Case Study of the Upper Blue Nile Basin (UBNB)

Eatemad Keshta[1,2], Doaa Amin[2], Ashraf M. ElMoustafa[1], Mohamed A. Gad[1]

[1]Irrigation and Hydraulics Department, Faculty of Engineering, Ain Shams University, Cairo 11517, Egypt
[2]Water Resources Research Institute (WRRI), National Water Research Center (NWRC), Ministry of Water Resources and Irrigation (MWRI), Qalyubia 13621, Egypt

*Correspondence to*: Eatemad Keshta (eatemad_hassan@nwrc.gov.eg; engeatemad4@yahoo.com)

**Abstract.**

Accurate simulation of precipitation over complex terrains such as the Upper Blue Nile Basin (UBNB) is essential for water resource management and climate impact assessments. The UBNB is characterized by complex terrain and convective precipitation systems that challenge the fine-scale climate simulation processes. This research aims to investigate the best precipitation parameterizations in the Regional Climate Model System (RegCM5) simulating different convective and large-scale schemes over the UBNB domain with 10 km resolution. The RegCM5 is driven by the fifth generation atmospheric reanalysis (ERA5) for the period of (2000-2009) using the hydrostatic dynamical core. The total precipitation simulations of the different calibration scenarios are assessed to select the optimal RegCM5 configuration over the UBNB. Results show that the model succeeds to capture the dominant spatiotemporal pattern of the precipitation, and the Emanuel scheme coupled with Nogherotto-Tompkins (NoTo) reduces the dominant wet bias in the precipitation simulation. The model highlights challenges in reproducing the UBNB's precipitation variability with a moderate to relatively good correlation of precipitation patterns from 0.46 to 0.77, where deficiency in capturing the large-scale circulations, especially the low-level circulations. The research recommended to focus on dynamics advancement, and exploring parameterization schemes that enhance the precipitation representation, such as the Planetary Boundary Layer (PBL) in Future.

## 1 Introduction

The seasonal rainfall over the Upper Blue Nile Basin (UBNB) is the main determinant of the variability in the entire River Nile basin hydrology, where there is a strong correlation between the fluctuations in both basins' flows (Conway and Hulme, 1993). A more reliable simulation of the UBNB climate, especially rainfall, can help in the water resources management of the riparian countries of the Nile basin. The UBNB is affected by the three Ethiopian climate seasons: Short Rain Season (February – May (FMAM)), Long Rain Season (June – September (JJAS)), and Winter Dry Season (October – January (ONDJ)) (Keshta, 2020). The various atmospheric systems that control the spatiotemporal variations of the UBNB rainfall seasons were comprehensively identified by (Camberlin and Philippon, 2002; Diro et al., 2011; Fekadu, 2015; Segele and Lamb, 2005). The FMAM weather pattern is derived by the interaction between the mid-latitude and tropical weather systems, while during the JJAS, the rainfall onset and distribution follow the Inter-Tropical Convergence Zone (ITCZ) oscillation and the anticyclones intensity of the southern hemisphere. The ONDJ is dominated by the northern hemisphere subtropical anticyclones and dry cool northeasterly monsoon. The JJAS represent the highest proportion of the total UBNB rainfall, with 70% to exceeding 75% (Mellander et al., 2013), where the average annual rainfall over the UBNB is around 1200 mm (Amin and Kotb, 2015).

The UBNB climate variability is affected by several global phenomena and mechanisms due to the ocean-atmosphere interaction. Among these mechanisms, the El-Nino Southern Oscillation (ENSO) and Indian Ocean Dipole (IOD) are the major drivers of the tropical climate (Coppola et al., 2012; Elsanabary and Gan, 2014; Siam and Eltahir, 2017). ENSO and IOD are phenomena relevant to the teleconnection of the rainfall variability (seasonal to interannual) controlled by the global Sea

Surface Temperatures (SSTs). (Abtew et al., 2009) found that the dry/wet years of the UBNB were linked to El Niño/La Niña events. El Nino disrupts the moisture transport into the basin reducing the rainfall, while La Nina enhances it promoting convective activities and causing heavy rainfall over the UBNB (Conway, 2000). A positive/negative IOD is associated with above/below-average rainfall over the UBNB due to the warm/cool water in the Indian Ocean near Africa (Elsanabary and Yew, 2015). (Elsanabary and Gan, 2015) explored the impact of the ENSO and IOD on the UBNB FMAM and JJAS rainy

seasons. They found that El Niño increases the FMAM rainfall and decreases the JJAS rainfall, while La Niña showed the opposite effect. However, during FMAM, the UBNB central part is unaffected by ENSO. The IOD has a wet effect on the FMAM and JJAS rainfall. At seasonal timescales, the ITCZ, as the main rain producing system, influences the spatiotemporal variability of rainfall over the UBNB (Tariku and Gan, 2018a; Zaroug et al., 2014). The ITCZ migration is governed by the Earth's tilt, so its effect varies with season due to the UBNB location in the northern hemisphere. During JJAS, the UBNB

captures high moisture from the Atlantic Ocean, released by the Ethiopian highlands, due to the developed subtropical high pressure systems with the blew wind from southwest to northeast together, which followed the ITCZ migration (Camberlin, 2009). Hence, the UBNB southwestern part is exposed to the westerly advective rains for a longer time than the northeast part. In ONDJ, the UBNB is located above the ITCZ, which migrates gradually to the southern hemisphere due to the wind blew from the north to south (Birhan et al., 2019). Therefore, the UBNB couldn't get sufficient precipitation. Moreover, the UBNB

has varied topography combining lowlands and high mountains; the Ethiopian Plateau, in which the elevation ranges from 2000 to more than 3500 m a.m.s.l (Shahin, 1985). This topographic altitude influences the fine-scale spatial distribution of the basin rainfall (Mohamed et al., 2005; Rientjes et al., 2013; Zeleke et al., 2013), since the mountain ranges generate local wind circulation patterns.

The Regional Climate Models (RCMs) can simulate climate over a region of interest at resolutions finer than the Global

Circulations Models (GCMs), providing more accurate information. RCMs performance is usually being assessed to investigate the climate characteristics, and studying the change of climate as well as land use impacts on the climate variables. A significant challenge in improving RCM performance in the area of interest lies in selecting the most appropriate physical parameterization schemes, developed for a specific climate condition and resolution. Hence, applying identical schemes produces different results not only in different regions but also in different seasons of the same region (Giorgi and Marinucci,

1996). It is demonstrated that the cumulus convection schemes (CCs) have a greater influence on the performance of RCM simulations than other schemes (Li et al., 2023) since CCs control the dynamics and the rainfall regimes variability.

In Africa, especially over Eastern Africa, the Nile basin, and the Sahel region, the versions of the Regional Climate Modelling system (RegCM) and Weather Research and Forecasting (WRF) have been commonly used for different climate applications. Over Eastern Africa, the performance of ten COordinated Regional climate Downscaling Experiment (CORDEX) RCMs

forced by European Centre for Medium-Range Weather Forecasts (ECMWF) Interim Re-Analysis (ERA-Interim) was assessed by (Endris et al., 2013) in simulating the rainfall. They found an overestimation over the Ethiopian highlands for all RCMs with relatively low spatial correlation; however, the ensembles' mean outperformed the individuals. (Tariku and Gan, 2018b) applied the WRF over the Nile basin, showing the same rainfall overestimation over the Blue Nile basin in the Ethiopian highlands and demonstrated this result as a predominance of a strong convective regime over the Indian Ocean. Despite this

wet bias, they concluded that the Kain-Fritsch CC (Kain, 2004) better simulated the rainfall over the entire basin. (Abdelwares et al., 2018) recommended another CC (Betts-Miller-Janjic scheme (BMJ) (Janjić, 1994)) when developing WRF over the Eastern Nile Basin. Focusing on the UBNB, they found that the combinations that used CCs of Kain-Fritsch and Grell 3D (Grell, 1993) highly overestimated rainfall compared to those that used BMJ, which captured the rainfall annual cycle with a small wet bias during the wet season.

Most studies using the RegCM over eastern Africa also found difficulties in reproducing the rainfall patterns correctly. (Segele et al., 2009) used the RegCM3 to simulate eastern Africa, reporting an overestimation of Ethiopia's precipitation when using the Grell and Emanuel (Emanuel, 1991) CCs. (Zeleke et al., 2016) evaluated the RegCM4 to simulate the precipitation of rain

seasons over the UBNB using the mixed CCs of Grell/Emanuel over land/ocean. Using the initial and boundary conditions of ERA-Interim they found that the precipitation was overestimated over the southwest and central regions and underestimated over the eastern region. Over West Africa, (Koné et al., 2018) found that a dry bias dominated the RegCM4 simulation, which was more pronounced using the CC of Tiedtke (Tiedtke, 1989) and recommended the Emanuel CC when using the RegCM4 with the land surface scheme of CLM4.5 (Community Land Model version 4.5 (Oleson et al., 2013)).

These selected schemes in previous works are limited due to the coarse model resolution that misses the finer local climate features. The fifth generation atmospheric reanalysis (ERA5) data provide advancements in the spatiotemporal resolution of ~31 km with 1 hour intervals. Compared with the ERA-Interim, ERA5 improves the vertical coverage, and tropospheric processes and tropical cyclones representation, enhancing the model's ability to simulate the precipitation in the deep tropics (Hoffmann et al., 2019). Therefore, in this study, we aim to evaluate the performance of the latest version of the Regional Climate Model (RegCM5 (Giorgi et al., 2023a)) over the UBNB using different combinations of convective and large-scale microphysics schemes. The main objective is to identify the most suitable configuration capable of accurately reproducing the observed precipitation characteristics and dominant seasonal variability of the basin. Establishing the optimal RegCM5 configuration over the UBNB will be considered a promising tool for more reliable regional applications, including climate change projection, seasonal forecasting, and the assessment of land use and land cover change impacts.

## 2 Model Description and Data

The RegCM5 is a freely available and flexible Regional Earth System model. It is the last version of the RCMs series developed at Abdus Salam International Centre for Theoretical Physics (ICTP), which was improved in collaboration with the Institute of Atmospheric Sciences and Climate of the National Research Council (ISAC-CNR) of Italy. A new dynamical core option (the non-hydrostatic core of the weather prediction model MOLOCH) has been added to this new version (Giorgi et al., 2023a). Now there are three options: hydrostatic, non-hydrostatic, and MOLOCH non-hydrostatic to be selected as a dynamical core option for a simulation. The RegCM5 parameterization set comprises various schemes such as the land surface, planetary boundary layer, sea surface flux, cumulus convective, microphysics, and radiation schemes. For the precipitation representation, there are five different cumulus convective schemes and three different microphysics schemes. In addition, mixed convective schemes over land and ocean can be used. More details on the model parameterization schemes can be found in (Giorgi et al., 2023b).

The model initial and boundary conditions are driven by atmospheric variables and SST from ERA5 hourly reanalysis data from the ECMWF (Hersbach et al., 2020) with 0.25º × 0.25º horizontal resolution for the period 2000-2009. For evaluation, observed daily rainfall data with a spatial resolution of 20 km × 20 km is obtained for the period of 2001-2009 from the Pre-Processor 7 (PP7). PP7 is a merge between gauge and satellite data blended in the Nile Forecasting System database (NFC., 2009) in the Egyptian Ministry of Water Resources and Irrigation.

## 3 Model Setup and Calibration

### 3.1 Domain

The domain for the RegCM5 simulation over the UBNB is selected to capture both local and large-scale atmospheric processes influencing the precipitation in the region. Figure 1a shows the UBNB domain, while Fig. 1b shows the UBNB topography. The domain has a 10 km resolution and extends from 5° S to 21° N latitude and 27° E to 53° E longitude (Fig. 1a), involving the different topographical features and a range of climate zones such as the Ethiopian Highlands, the Western Indian Ocean, the Arabian Peninsula, and the Southern Red Sea. The model utilizes 18 vertical sigma levels and a top at 50 hpa. Table 1 involves the details of the domain extent. The selected domain is large enough to ensure more reliable simulations of the main

climate characteristics, among which the moisture transport from the main sources for the region's precipitation dynamics, like the East African Monsoon and the Indian Ocean (Endris et al., 2013).

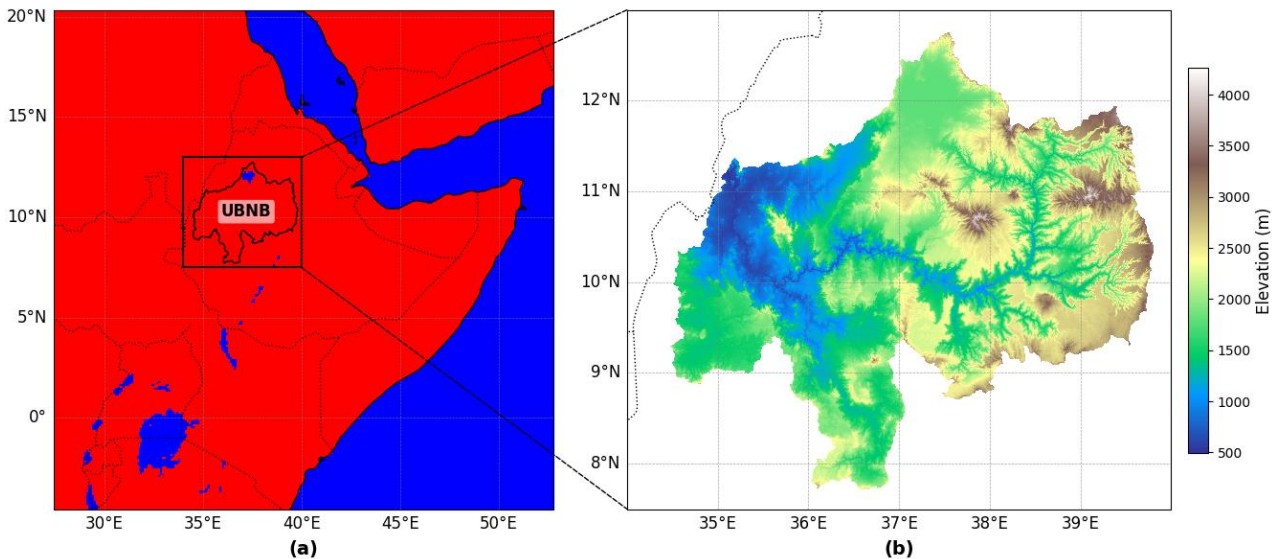

**Figure 1: The extent of the UBNB domain. [(a) Land Mask of the UBNB domain; (red for land and blue for water), and (b) UBNB topography; (unit: m)].**

**Table 1. The domain extent**

| | |
|---|---|
| Longitude | 27° E - 53° E |
| Latitude | 05° S - 21° N |
| Nesting Ratio | 1 : 3 |
| Resolution | 10 km |
| No. of grid cells | 280 × 280 |
| Vertical layers | 18 vertical sigma levels |
| Top pressure | 50 hPa |

### 3.2 Physics Parametrization

The hydrostatic dynamic core (Giorgi et al., 1993) is used for the RegCM5 configuration in the different numerical
experiments. To optimize the number of trials due to the high computational cost, the selection of the parameterization schemes representing the precipitation is divided into two parts to be tested over the UBNB, where the second part has resulted from the initial evaluation of the first one, as follows:

1. First, four Cumulus Convective (CC) schemes are tested: Grell (closure of Fritsch-Chappell), Emanuel, Tiedtke, and Kain-Fritsch schemes, to represent the convective precipitation over land. Over the ocean, only the Emanuel scheme is selected
as mixed convection with all four CCs over land. The SUB-grid EXplicit (SUBEX) (Pal et al., 2000) is used for large-scale (resolvable, or non-convective) precipitation. The SUBEX is selected as a resolved-scale cloud physics option, which is usually used in the earlier RegCM versions over the UBNB or around the basin, such as Ethiopia, East Africa, and West Africa (Endris et al., 2013; Koné et al., 2018; Nikulin et al., 2012; Segele et al., 2009; Zeleke et al., 2016).

2. Second, after the initial evaluation of the results, a high overestimation of the precipitation is noticed, especially in the
scenario that used the Grell. As a result, the Grell is excluded, and a decision was made to enhance the choice of schemes that can also affect the precipitation simulation. Hence, a new set of simulations was conducted by using Nogherotto-Tompkins (NoTo; (Nogherotto et al., 2016)) microphysics scheme, which treats the mixed-phase clouds, removing the oversimulation of the upper level cloud characteristics of the SUBEX scheme. Over East Africa, (Gudoshava and Semazzi, 2019) revealed that the NoTo generally reduces the overestimation of CCs; however, they recommended the SUBEX with
the Grell (Fritsch-Chappell closure). In addition, (Kalmár et al., 2021) tested different resolved-scale cloud microphysics

schemes over a mountainous region in eastern-central Europe. They found the SUBEX overestimated the high intensity tail of the observed precipitation, while the NoTo reproduced it better whatever the type of the dynamical core (hydrostatic or non-hydrostatic). Therefore, in this research, the NoTo is added as another option for large-scale precipitation.

The planetary boundary layer (PBL) scheme developed by (Holtslag et al., 1990) is used to represent the vertical interaction

between the surface (land or ocean) and the atmosphere. In addition, the ocean flux scheme developed by (Zeng et al., 1998) is used and the rapid radiation transfer scheme (RRTM) is used as the radiation scheme. Finally, the CLM4.5 is used for the land surface representation. These schemes are chosen based on recommendations provided in previous studies conducted near the UBNB as the choice of SUBEX (above). It should be noted that in all convection schemes, the default parameter values are used. Table 2 summarizes the selected parameterization schemes for the different seven calibration scenarios.

**Table 2. Combination of physical parameterization schemes selected for calibration**

| Scenario No. | Land CC | Ocean CC | Microphysics | PBL | Ocean Surface Flux | Radiation | Land Surface | Scenario Name |
|---|---|---|---|---|---|---|---|---|
| S1 | Grell | | SUBEX | | | | | Grell_SUBEX |
| S2 | Emanuel | | | | | | | Emanuel_SUBEX |
| S3 | Tiedtke | | | | | | | Tiedtke_SUBEX |
| S4 | Kain-Fritsch | Emanuel | | Holtslag | Zeng | RRTM | CLM4.5 | Kain-Fritsch_SUBEX |
| S5 | Emanuel | | NoTo | | | | | Emanuel_NoTo |
| S6 | Tiedtke | | | | | | | Tiedtke_NoTo |
| S7 | Kain-Fritsch | | | | | | | Kain-Fritsch_NoTo |

**3.3 Calibration Methodology**

Some statistical criteria are used for the performance evaluation of the different physical parametrizations of RegCM5 over the UBNB. Two statistical criteria, relative bias (Bias%), Eq. (1), and Root Mean Squared Error to observation Standard Deviation Ratio (RSR), Eq. (2) (Moriasi et al., 2007), are calculated on a monthly time series basis as an error indication. The

monthly mean spatial precipitation over the UBNB is first computed, and then this basin-averaged monthly time series is used to calculate the Bias% and RSR between simulated and observed precipitation. According to (Moriasi et al., 2007), model performance is considered satisfactory when RSR ≤ 0.70.

$$Bias\% = \left[ \frac{\sum_{t=1}^{n}\left(Y_t^{sim} - Y_t^{obs}\right)*100}{\sum_{t=1}^{n}\left(Y_t^{obs}\right)} \right], \tag{1}$$

$$RSR = \left[ \frac{\sqrt{\sum_{t=1}^{n}\left(Y_t^{sim} - Y_t^{obs}\right)^2}}{\sqrt{\sum_{t=1}^{n}\left(Y_t^{obs} - \overline{Y^{obs}}\right)^2}} \right], \tag{2}$$

where $n$ is the number of observations, $Y_t^{sim}$, $Y_t^{obs}$ are the simulated and observed precipitation at time $t$, respectively, and $\overline{Y^{obs}}$ is the mean of the precipitation observations during the calibration period.

To check the annual cycle and spatial distribution, some additional criteria are computed for the three climate seasons FMAM, JJAS, and ONDJ. The correlation coefficient in time is calculated for the three seasons to measure the strength of a linear association between the simulations and observation patterns for each grid point. The correlation coefficient ($R^2$) is

given by Eq. (3).

$$R^2 = \frac{\sum_{t=1}^{n}\left(Y_t^{obs} - \overline{Y^{obs}}\right)\left(Y_t^{sim} - \overline{Y^{sim}}\right)}{\sigma^{obs}\,\sigma^{sim}}, \tag{3}$$

where $\overline{Y^{sim}}$ is the mean of the simulated, and $\sigma^{obs}$ and $\sigma^{sim}$ are the standard deviations of the observed and the simulated precipitation.

Finally, the Brier Score (BS) and Significance Score (SS) (Brier, 1950; Fraedrich and Leslie, 1987), are estimated to assess the probability density function (PDF) for the simulated and observed daily data during the three seasons. A wet day is defined as a day with precipitation greater than 1 mm/day. This threshold was applied to exclude trace amounts and ensure consistency when evaluating precipitation intensity and frequency. The BS represents the mean square error of the probability, and SS represents the smallest cumulative probability of the observation and simulation distribution in each equal sequence of values. BS and SS are given as follows:

$$BS = \frac{1}{N}\sum_{i=1}^{N}(P_i^{sim} - P_i^{obs})^2 , \tag{4}$$

$$SS = \sum_{i=1}^{N} Minimum(P_i^{sim}.P_i^{obs}) , \tag{5}$$

where $N$ is the number of intervals, $P_i^{sim}$ is the probability density value of the simulated precipitation at the interval $i$, and $P_i^{obs}$ is the probability density value of the observed precipitation at the interval $i$. The smaller/larger BS/SS indicates the ability of the RegCM5 scheme to simulate the probability density distribution.

## 4 Results and Discussion

The results are analyzed for the period of 2001-2009 since the year 2000 was considered as a spin-up period. The simulated precipitation is compared to the observed PP7 data.

### 4.1 Error-based Evaluation

To evaluate the different RegCM5 configurations over the UBNB, Fig. 2 and Table 3 show the evaluation results of the simulated precipitation for the seven numerical experiments. Figure 2a shows the mean monthly precipitation, which represents the mean annual cycle over the UBNB. The annual mean of the PP7 is about 1232 mm. Most of the RegCM5 simulations follow the observed precipitation pattern, but there is a high overestimation, especially for S1, which uses the Grell scheme. It is also noticed that the overestimation is reduced when changing the microphysics scheme from SUBEX to NoTo, indicating that SUBEX overestimates the large-scale precipitation over the UBNB. For NoTo scenarios, both Emanuel and Tiedtke (S5 and S6) are closer to the PP7 than Kain-Fritsch (S7), especially at the onset of the rainy seasons (FMAM & JJAS). The NoTo not only corrects the overestimation but also reduces the difference between the CCs.

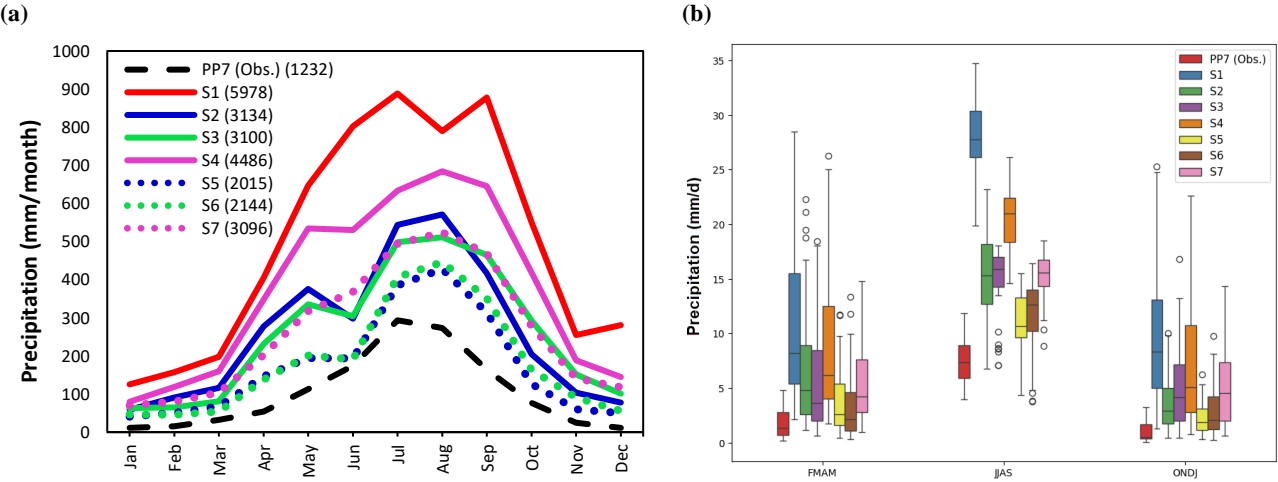

**Figure 2. Simulations of precipitation over UBNB; (a) the mean monthly, and (b) boxplots of the monthly mean during rain and dry seasons.**

To analyze the variation of the three climate seasons (FMAM, JJAS, and ONDJ) over the UBNB, the monthly mean precipitation boxplots are presented in Fig. 2b. It is noticed that NoTo (S5, S6, and S7) succeeded to reduce the precipitation range. In addition, it captures the low rainfall values in the wet season (JJAS). S5 & S6 boxplots are also closer to the PP7 boxplots than the other scenarios. Table 3 reports the computed statistical criteria that investigate the error of the model

simulations of the mean areal precipitation over the basin. It is found that the S5 has the lowest and best values of the Bias% and RSR. The RSR is out of the accepted range ($0 \leq$ RSR $\leq 0.7$); however, as shown in Fig. 3a, all the experiments successfully capture the dominant temporal pattern of rainfall variability. For the considerable wet bias, previous studies have demonstrated that it is important for the RCMs to perform well in capturing the dominant spatiotemporal pattern of the climate variability than the absolute values of the bias. For example, (Koné et al., 2018) tested the sensitivity of the RegCM4 to different convective schemes over West Africa. They found that Emanuel succeeded to reduce the dominant dry bias that ranged between 26 to 43 % over different regions in West Africa. Hence, bias correction is generally required before using the simulated variables for any hydrological impact or application studies (Haerter et al., 2011; Sippel et al., 2016; Teutschbein and Seibert, 2012). For example, (Osman et al., 2021) tested the WRF model sensitivity to get its optimum configuration over the Eastern Nile. Their results showed highly underestimated precipitation over the UBNB; therefore, they corrected the simulation using a bias correction method before applying it to the hydrological model.

**Table 3. Error-based statistical criteria**

| Scenario | Bias% | RSR |
|----------|-------|------|
| S1 | 339 | 4.60 |
| S2 | 153 | 2.10 |
| S3 | 149 | 2.00 |
| S4 | 261 | 3.25 |
| S5 | 66 | 1.00 |
| S6 | 76 | 1.20 |
| S7 | 155 | 1.90 |

**4.2 Spatiotemporal Evaluation**

The performance of the experiments S5, S6, and S7, which use the NoTo microphysics scheme, is evaluated by analyzing the spatial pattern and the intra-annual variability for the three seasons. Figure 3 shows the Bias% with respect to PP7 observation data. The bias distribution indicates that the model tends to overestimate precipitation in the central and southern mountainous regions of the basin, where high rainfall is typically observed. This positive bias is particularly pronounced in the Kain-Fritsch scenario. In contrast, the model also exhibits substantial positive bias over the eastern and southwestern regions, which are semiarid zones that generally receive low precipitation compared to the central and southern parts. The bias distribution shows spatial discontinuities in simulated precipitation, especially in the northeast. These discontinuities are mainly attributed to the strong topographic gradients and land–atmosphere interactions over the northeastern highlands of the basin, where local convective triggering is highly sensitive to terrain-induced uplift and land surface heterogeneity. Emanuel, which has the lowest overestimation, underestimated the precipitation in the western region in JJAS with a negative bias of about 10%. In the FMAM season, Tiedtke has a slightly lower overestimation than Emanuel.

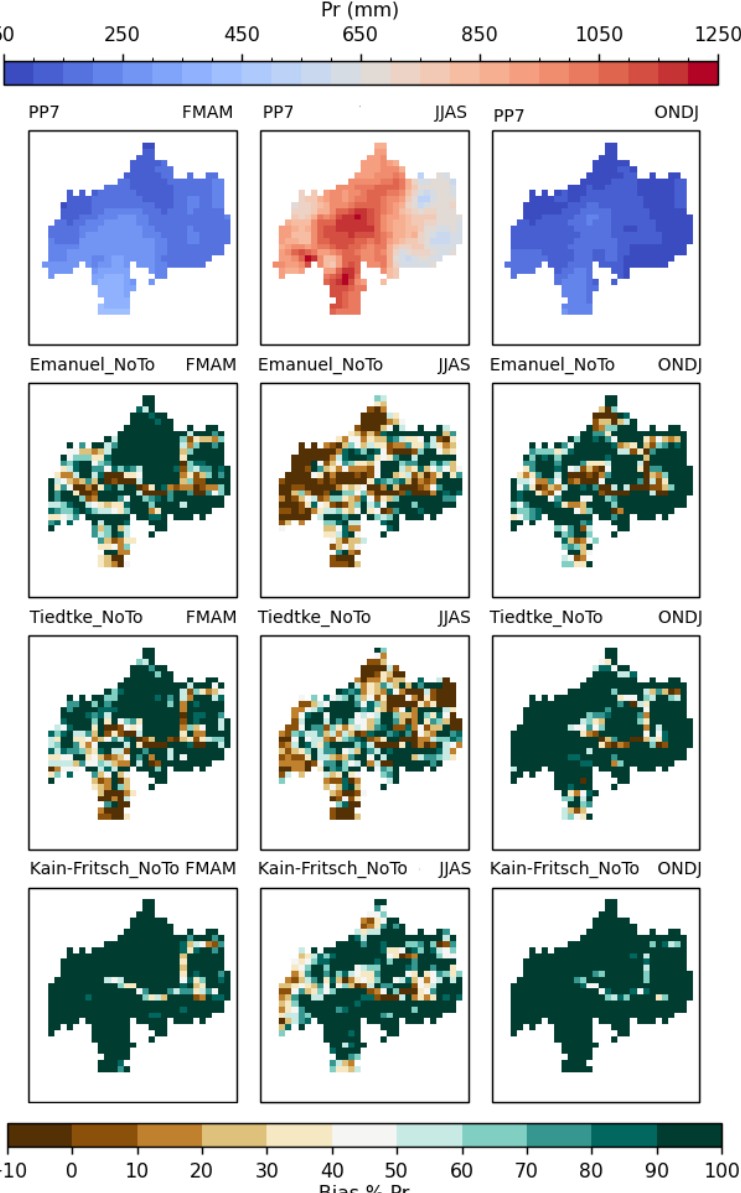

**Figure 3.** Seasonal mean precipitation percent bias (Bias%) with respect to the PP7 observation data (at the top panel) over the UBNB during the three seasons (FMAM, JJAS, and ONDJ) at (left, middle, and right) columns.

Figure 4 represents the spatial variation of the correlation coefficient between the monthly mean time series of the simulated precipitation of S5, S6, and S7 and the PP7 observed precipitation during the three seasons. The correlation ranges from 0.46 to 0.77, showing a moderate to relatively good relationship between the simulated and observed precipitation. This indicates that the model captures the temporal variability reasonably well, but the model configurations still need improvement.

All experiments show similar correlation performance with close values across the basin; however, the spatial patterns of correlation provide critical insight into areas where the model exhibits deficiencies. For the FMAM season, weak to moderate correlations dominate most of the basin, and the eastern regions exhibit a lack of strong correlation. In the east of UBNB, short rains during FMAM exhibit a low correlation. In JJAS, the model shows higher correlations in the west, while a negative correlation over the southwestern part.

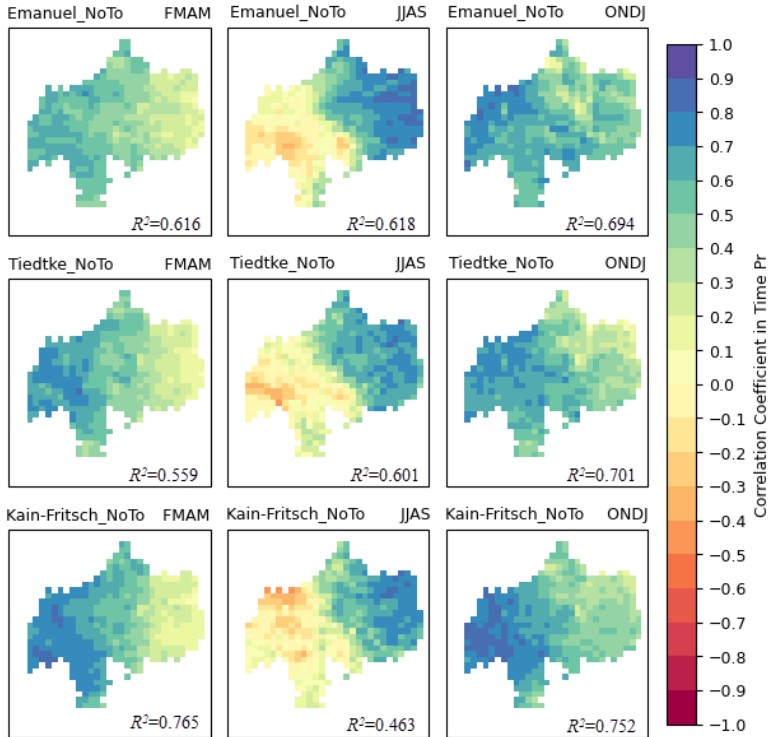

240

**Figure 4.** Correlation coefficient with respect to the PP7 observation data over the UBNB during the three seasons (FMAM, JJAS, and ONDJ) at (left, middle, and right) columns.

The PDFs are analyzed to assess the daily precipitation characteristics simulated in the experiments S5, S6, S7 and observed by PP7 during the FMAM, JJAS, and ONDJ (Fig. 5). The simulations couldn't capture the PDF of PP7, especially the low and

245 mid precipitation intensity, which is very clear during the JJAS (Fig. 5b). However, the PDF of Emanuel_NoTo (S5) is closer to the PP7 than the other simulations. Tiedtke_NoTo (S6) slightly better represents the observed distribution of the daily precipitation during the FMAM (Fig. 5a).

(a)

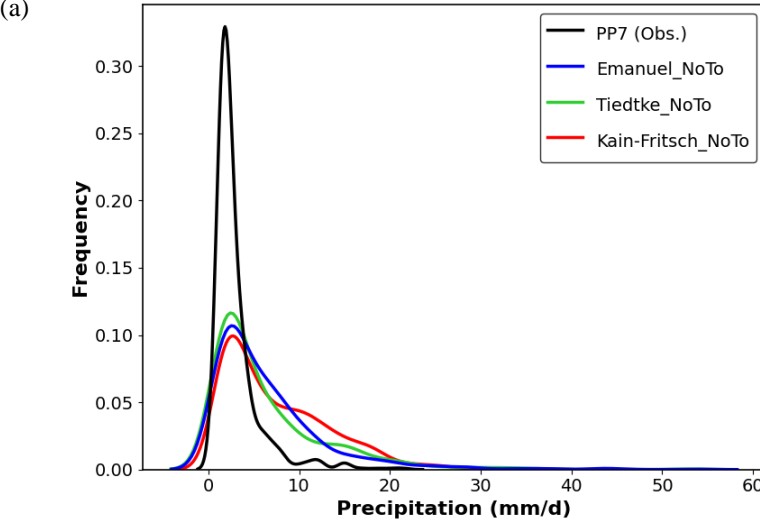

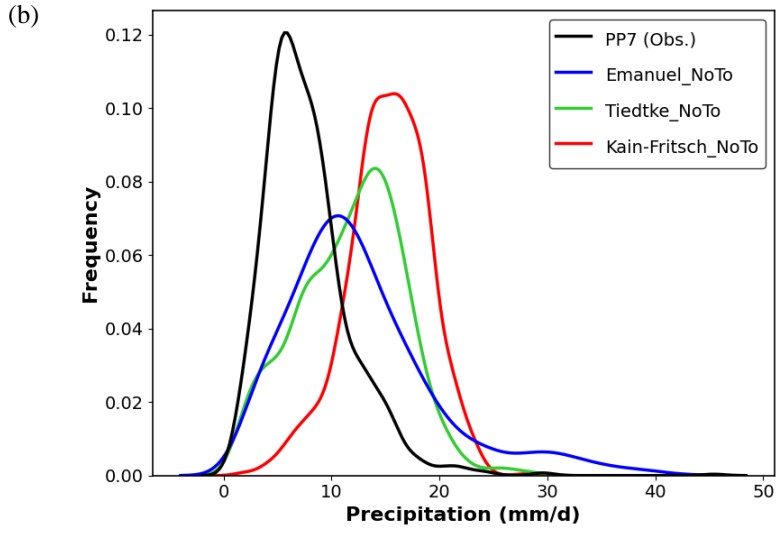

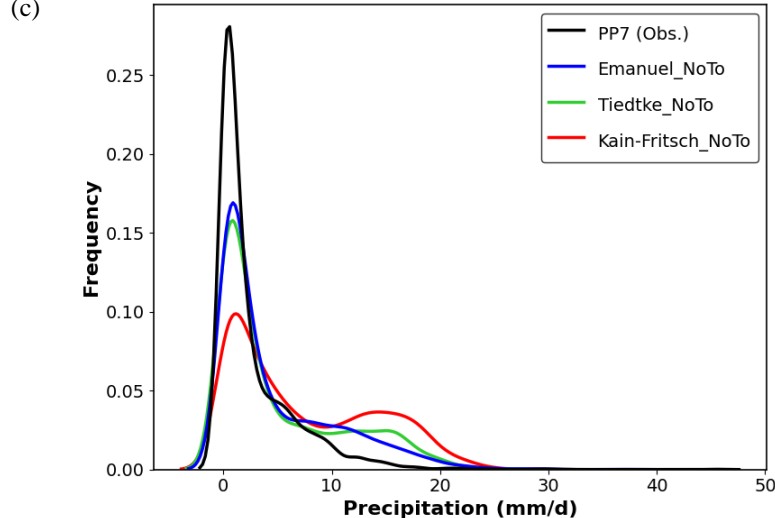

**Figure 5. The PDF of the daily precipitation over the UBNB during seasons; (a) FMAM), (b) JJAS, and (c) ONDJ.**

The BS and SS scores are reported in Table 4 to evaluate the PDFs. Emanuel_NoTo (S5) has the lowest BS and the highest
250 SS (best results) during JJAS and ONDJ, while for FMAM, the best BS and SS found when using Tidetke_NoTo (S6). This
demonstrates the visualization interpretation from Fig. 5. The intensity of precipitation events is also influenced by the
positioning and dynamics of upper-level jets and troughs (Qi et al., 2023). Therefore, RegCM5 may face challenges in
accurately representing these complex interactions due to limitations in parameterizations and resolution. Refining the spatial
scale increases the natural variability of the precipitation challenging the detection of forced signals (Giorgi, 2002). Hence, at
255 such high resolution simulation, hydrostatic dynamics may struggle to correctly parameterize interactions between local and
large-scale circulations. It should also be noted that the observed data may be affected by significant uncertainties. The PP7 is
a merge between gauges, collected from data summaries provided by the World Meteorological Organization (WMO), and
satellite data. The reported records of rain gauges that cover the UBNB are not error free, since not all the zero readings
occurred, but there is a possibility that rainfall occurred but was not reported (Keshta et al., 2019).

260 **Table 4. Scores of PDFs of the simulated daily precipitation over the UBNB**

| Scenario | BS | | | SS | | |
|---|---|---|---|---|---|---|
| | FMAM | JJAS | ONDJ | FMAM | JJAS | ONDJ |
| **Emanuel_NoTo** | 0.0010 | 0.0007 | 0.0008 | 1.17 | 1.35 | 1.57 |
| **Tiedtke_NoTo** | 0.0006 | 0.0010 | 0.0009 | 1.31 | 1.25 | 1.50 |
| **Kain-Fritsch_NoTo** | 0.0017 | 0.0024 | 0.0017 | 0.96 | 0.64 | 1.26 |

Overall, Emanuel with NoTo succeeded to simulate the UBNB precipitation, especially the JJAS, which represents ~70% of the total annual precipitation over the UBNB. However, from the weak correlation in some parts of the UBNB; eastern/southwestern during FMAM/JJAS, reflects the deficiency of the model to simulate the large-scale circulation associated with the rainfall generation during these seasons. During the FMAM rains over Ethiopia, the large-scale convection in the lower troposphere is fostered by the downward bent of the subtropical westerly jet (SWJ) at upper levels (Zeleke et al., 2016). Similarly, (Fekadu, 2015) highlighted that the interaction of the SWJ with deep troughs in the easterly flow enhances upward motion and moisture convergence, which is critical for rain production during FMAM. For the major long rains during JJAS, the southwestern UBNB is exposed to westerly rains for a longer time than the eastern and northeast parts (Mellander et al., 2013) benefiting from lower tropospheric southwesterlies from the Atlantic (Nicholson, 2017) due to the windward side of the Ethiopian Highlands. These westerly rains are attributed to the large-scale circulation, such as the tropical easterly jet (TEJ) and the Eastern Africa Low-Level Jet (EALLJ). TEJ and EALLJ, with the quasi-permanent high-pressure systems over the South Atlantic and South Indian Ocean, together affect the quality of the JJAS rain season (Camberlin, 2009; Mohamed et al., 2005). The formation and movement of these systems, along with their interactions with local topography and atmospheric conditions, are essential in driving precipitation patterns. For example, large-scale features like the TEJ and shifts in the position of troughs can create instabilities that result in significant rainfall (Yin et al., 2023).

Therefore, the large-scale circulation analysis has been tested for the best performing experiment configuration (Emanuel + NoTo) by comparing the model simulation to the ERA5 reanalysis. Figure 6 shows the model's upper- and lower-level winds compared to the ERA5 reanalysis over the UBNB at 200 hPa and 850 hPa, respectively, during FMAM. The 200 hPa circulation simulated by RegCM5 closely matches ERA5, reproducing the main SWJ structure. However, at 850 hPa, the model overestimates the inflow to the UBNB from the Indian Ocean while underestimating the westerly component originating from the Atlantic Ocean. This imbalance in the simulated low-level circulation likely alters the moisture convergence pattern over the basin. The over-intensified easterly-to-southeasterly inflow from the Indian Ocean enhances moisture transport toward the eastern and northern highlands, regions that are typically semi-arid during FMAM. Consequently, this moisture influx can promote excessive convective activity and lead to the positive precipitation bias observed in these regions. Conversely, the underrepresentation of the westerly component reduces the west-to-east advection of moist air masses from the Atlantic and Congo Basin, which normally contribute to the realistic spatial distribution of rainfall. The weakened westerly inflow, therefore, diminishes the dynamic balance between the two moisture sources, resulting in misplaced convergence zones and reduced rainfall correlation with observations in the eastern and northern UBNB.

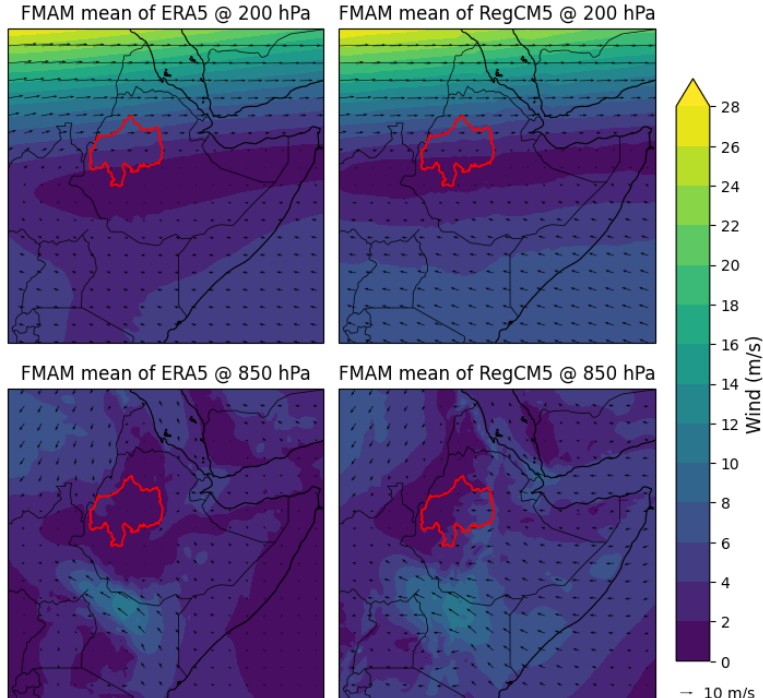

**Figure 6. FMAM mean wind (m/s; shading) and wind vectors at 200 hPa (upper panels) and 850 hPa (lower panels) from ERA5 (left) and RegCM5 (right) over the UBNB.**

For the JJAS, Fig. 7 shows the model's upper- and lower-level winds compared to the ERA5 reanalysis over the UBNB at 200 hPa and 850 hPa, respectively. At 200 hPa, the large-scale upper-tropospheric flow is broadly similar between ERA5 and RegCM5, indicating that the model captures the TEJ feature during JJAS. Small differences in jet latitude/intensity are present (the model shows slightly localized maxima northeast of the basin), but overall, the upper-level circulation is reproduced reasonably well. At 850 hPa, it is noticed that the model produces an additional northerly/northeastward inflow into the basin that does not exist in ERA5, and at the same time, some of the strong southwesterly flow shown in ERA5 (which brings Atlantic/Congo moisture into the southern and southwestern UBNB) is reduced or displaced. This anomalous northerly contribution in the model reduces the relative importance of the southwesterly moisture supply to the southern and southwestern regions. Because JJAS rainfall in those parts of the basin depends strongly on the low-level southwesterly moisture inflow and convergence on the windward side of the highlands, the model's altered low-level circulation plausibly explains the noticed weak correlation between model and observation.

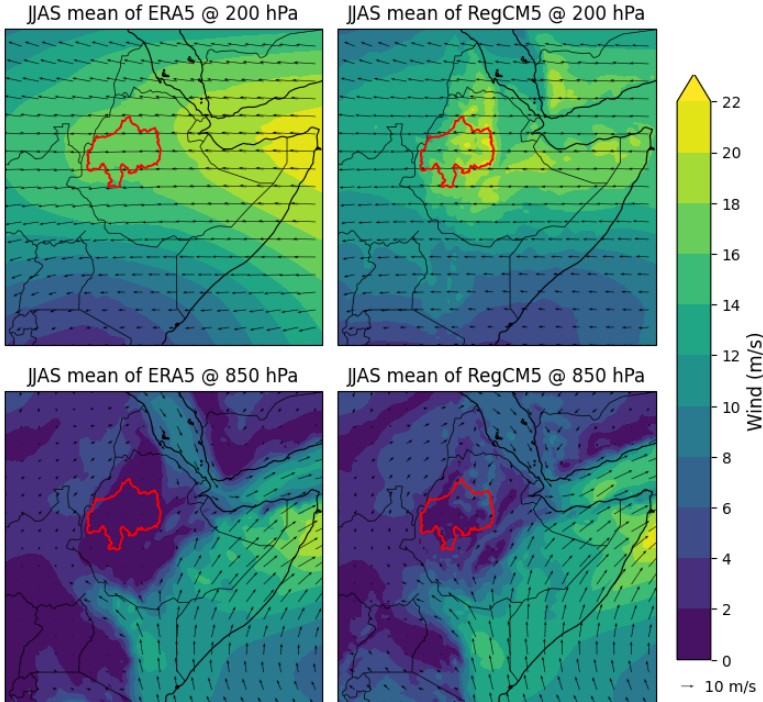

**Figure 7. JJAS mean wind (m/s; shading) and wind vectors at 200 hPa (upper panels) and 850 hPa (lower panels) from ERA5 (left) and RegCM5 (right) over the UBNB.**

The good performance of the Emanuel + NoTo configuration likely reflects complementary strengths of the two parameterizations. Emanuel's mass-flux, parcel-based closure is sensitive to environmental humidity and entrainment/detrainment and therefore can better represent convective triggering and organization in orographically influenced regimes (Emanuel, 1991). The NoTo microphysics provides a multiple-phase, prognostic treatment of cloud liquid, ice, rain and snow and yields a more realistic mixed-phase cloud structure and stratiform precipitation (Nogherotto et al., 2016). Together, the Emanuel closure (improved convective moistening and triggering) and NoTo microphysics (improved ice-phase and stratiform processes) change the partitioning between convective and large-scale precipitation and improve the vertical cloud profile, which is especially important in the high-relief, mixed convective/stratiform environment of the UBNB.

However, the model configuration requires enhancement to reduce the bias in the other seasons (FMAM and ONDJ) and generally improve the spatiotemporal pattern over the UBNB. Using the new dynamic core option, MOLOCH non-hydrostatic involved in the RegCM5, may enhance capturing the observed rainfall variability. The non-hydrostatic dynamic core improves the representation of mesoscale convective systems and tropical storms (Giorgi, 2019). Thus, it can resolve fine-scale atmospheric processes circulations in regions with complex terrain and convective precipitation systems such as the UBNB. (Silué et al., 2024) found that using MOLOCH non-hydrostatic improves the simulation of precipitation intensity, diurnal cycles, and the representation of mesoscale convective systems compared to the hydrostatic core. They also found that using the PBL scheme of the University of Washington (UW) (Bretherton et al., 2004) instead of its counterpart, Holtslag, showed a better representation of boundary-layer dynamics and vertical mixing. UW employs higher-order turbulence parameterizations and showed outperformance in capturing vertical profiles of temperature, humidity, and wind, leading to improved precipitation simulations during the rainy season (JJAS) over West Africa. Counting for such an update can also improve the precipitation simulation over the UBNB.

## 5 Conclusions

In this research, we investigated the performance of the RegCM5 (hydrostatic dynamical core), using the advancement of the spatiotemporal resolution of the ERA5, to simulate the spatiotemporal variability of the precipitation over the UBNB for the wet and dry seasons. The model captures the general pattern of the observed rainfall, although it is overestimated compared to

the PP7 observation data. The non-convective precipitation is highly overestimated. The NoTo microphysics scheme outperforms the SUBEX in representing the non-convective precipitation and reduces the biases in the simulated total precipitation. Exploring the better performance of NoTo than SUBEX, which has widely been used in earlier studies, especially over East Africa, is considered a novelty of the research. Emanuel CC over land demonstrates a relatively accurate representation of convective precipitation, which dominates rainfall in UBNB, especially during the long rain season (JJAS).

During JJAS, the model spatially captures the high rainfall locations in central and southern regions; however, it underestimated the western UBNB with a negative bias of up to 10%. Due to the high overestimation, the model couldn't capture the low and mid intensities, which is clearly noticed in the daily PDFs.

Comparing the temporal correlation between the simulations and observation data spatially provides critical insight into areas where the model exhibits deficiencies. The model exhibits limited capability in reproducing the spatial rainfall distribution over the eastern (southwestern) parts of the basin during FMAM (JJAS), leading to weak or negative correlations with the observed datasets. The large-scale circulation analysis reveals that these deficiencies are linked to misrepresented low-level wind structures. During FMAM, the RegCM5 simulation overestimates the easterly inflow from the Indian Ocean while underrepresenting the westerly contribution from the Atlantic Ocean, thereby distorting the moisture convergence and increasing the rainfall over the basin. Conversely, in JJAS, the model generates an anomalous northerly component and weakens the southwesterly monsoon flow responsible for transporting moist air from the Atlantic and Congo regions toward the Ethiopian highlands. This deviation in the simulated low-level circulation likely alters the moisture transport pathways and weakens the rainfall–circulation coupling over the UBNB.

In conclusion, the model reasonably succeeded to simulate the dominant spatiotemporal annual pattern of the precipitation over the UBNB using Emanuel with NoTo, since it reproduces the UBNB mean annual close to the PP7 observation data with a wet bias. The success of the model in capturing the spatial variability of the JJAS with a slight dry bias, which represents the highest proportion of the UBNB annual precipitation (~70%), is promising for future enhancement. To enhance the model configuration, we recommend using the new dynamic core option of MOLOCH non-hydrostatic that can play a role in resolving these upper-atmosphere processes, reducing biases in simulating the precipitation. In addition, we recommend to test more physical parameterizations of the RegCM5 that affect the precipitation simulation (e.g., PBL schemes).

**Code and Data availability:** The RegCM5 code is available from the project website: https://github.com/graziano-giuliani/RegCM/tree/5.0.0. The RegCM5 used to produce the results used in this paper is archived at Zenodo (Giorgi et al., 2023a). The model input data is available at http://clima-dods.ictp.it/regcm4. The ERA5 reanalysis data is available at https://cds.climate.copernicus.eu/datasets/reanalysis-era5-pressure-levels for the initial and boundary conditions, and https://cds.climate.copernicus.eu/datasets/reanalysis-era5-single-levels for the SST data. The observed precipitation data (PP7) used in this research are provided by the Egyptian Ministry of Water Resources and Irrigation (WMRI) and cannot be shared publicly due to data restrictions. The results and codes used to produce the plots for all the simulations presented in this paper are uploaded at Zenodo (Keshta, 2025) (DOI: https://doi.org/10.5281/zenodo.14864919).

**Author contribution:** EK was responsible for software development, formal analysis, investigation, data curation, writing the original draft, and visualization. DA provided the resources. DA, AME, and MAG supervised the research. All authors contributed equally in conceptualization, methodology, validation, and writing - review and editing.

**Competing interests:** The authors declare that they have no conflict of interest.

**Acknowledgements:** The authors would like to express their sincere thanks to the Nile Forecast Center (NFC), Ministry of Water Resources and Irrigation (MWRI), Egypt for providing the rainfall data of the NFS.

This paper is based upon work supported by Science, Technology & Innovation Funding Authority (STDF) under the 2nd Post Graduate Support Grant (PGSGII).

**Financial Support:** This research is supported through a project entitled "Projection of Stored Water Upstream Dams on Microclimate" (No. 48620), which is funded by STDF in Egypt.

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
