# Peer review of "Comparison of Precipitation Parameterizations in Regional Climate Model (RegCM5): A Case Study of the Upper Blue Nile Basin (UBNB)"

_EGUsphere, 2025_

## Author Comment (AC1)

**Reply on Referee #1 Comments**

Review: "Optimizing Precipitation Parameterizations in Regional Climate Model (RegCM5): A Case Study of the Upper Blue Nile Basin (UBNB)"

A set of 7 numerical experiments using Regional Climate Model version 5 (RegCM5) was conducted to understand the better precipitation parametrization to reproduce the monthly and daily precipitation over the upper blue Nile basin (UBNB). The numerical experiments combine different cumulus convection schemes (over the land the Kain-Fritsch, or Grell, or Emanuel, or Tiedke, or Emanuel are used, combined with Emanuel over the sea) and large-scale precipitation schemes (SUBEX and NoTo). Simulations were driven by ERA5 reanalysis and PP7 precipitation data was used as observational reference. For the comparison between simulated and observed precipitation it is used some statistical indices. The main conclusion is that predominates a general overestimation of the simulated rainfall in most seasons of the year, with an acceptable performance being obtained by the combination of Emanuel for cumulus convection and NoTo for large-scale precipitation schemes. The analysis of simulation performance in the region is a relevant topic with large potential to add new knowledge related to models' physical parameterizations behavior in the tropics. However, as highlighted in the major points, there are some statements/interpretations that need to be clarified/better founded before the acceptance of the manuscript.

**Major points**

The title does not correspond to the analysis presented in the manuscript. The
authors did not conduct any "optimization" in the model since they used the
default code and parameters. The manuscript only presents a comparison of
different schemes to represent the moist deep convection and large scale
precipitation. One suggestion of title is:

"Comparison of precipitation parametrizations in Regional Climate Model (RegCM5): A case study for Upper Blue Nile Basin (UBNB)"

**Done. Thanks.**

2. In L226-240 (L287-289 of conclusions) the authors attributed the simulated precipitation errors to the difficulty of the simulations to reproduce important features of large scale circulation in the region. However, no figures/tables and their interpretations are presented to justify the claims which are only based on the previous literature. The authors need to improve this part of the manuscript

**by presenting/analyzing how the experiments simulate the mentioned circulation features.**

We appreciate the reviewer's valuable comment. In the revised manuscript, we have added a new analysis of the large-scale circulation to support our discussion. Figures 6 and 7 now present the mean wind at 200 hPa and 850 hPa from both ERA5 and RegCM5 for the FMAM and JJAS seasons. The discussion part in section 4.2 (L275-312) interprets how discrepancies in upper- and lower-level circulation patterns, such as the over-intensified easterly inflow from the Indian Ocean during FMAM and the weakened southwesterly monsoon flow during JJAS, affect the simulated rainfall distribution and bias over the UBNB. Correspondingly, the conclusion has been updated to reflect these findings in L339-347

3. (L264-271) After identifying a considerable overestimation of rainfall by the simulations, the authors present several arguments that could have provided more realistic simulations. My question is: Why didn't the authors use these recommendations in their numerical experiments? In particular, the use of a hydrostatic dynamics at 10 km resolution, which is considered the interface (or limit) of validity for this approximation, is highly questionable. Why did not the authors use the MOLOCH that is non-hydrostatic and makes the RegCM5 code very fast? The authors need to clearly justify their decision to use the hydrostatic approximation core.

We thank the reviewer for this important point. We selected the hydrostatic dynamical core for the experiments for the following reasons:

- 1. In earlier studies that relied on ERA-Interim reanalysis, achieving such high resolution using the hydrostatic dynamical core typically required double nesting (e.g., downscaling from around 50 km to 12–10 km) (Prein et al., 2016; Torma et al., 2015). Therefore, the current higher spatial and temporal resolution of ERA5 encourages us to use it for direct simulations at 10 km (the interface) at once.
- 2. In (Giorgi et al., 2023), that describes the release and development of the fifth RegCM, simulations using the new non-hydrostatic core have been tested, exhibiting better performance than the hydrostatic core, however; they stated that further optimization work in under way to fully test the model in different climate settings and reduce current biases.

These reasons made us decide to use the hydrostatic core first. In addition, despite identifying the considerable bias, the model successfully captures the dominant pattern of the precipitation variability over the basin. This is more important than the

absolute values for certain applications, since it is recommended to do a bias correction for the climate variables simulated from the RCMs before it is used in applications such as hydrological applications. For example, (Osman et al., 2021) tested the WRF model sensitivity towards different microphysics and Planetary Boundary Layer (PBL) parameterization schemes over the Eastern Nile. The precipitation was highly underestimated over the UBNB; therefore, they corrected the simulation using a bias correction method before applying it to the hydrological model.

Finally, we would like to clarify our decision to stop at this situation and publish our results as such experiments using the hydrostatic core, despite the high resolution of ERA5 reanalysis as an input to RCMs, there will be a considerable bias at the interface resolution of 10km, and it is recommended to test the new non-hydrostatic option. We have already conducted this recommendation to be published as a new work.

4. Figure 4 and 5 are redundant since both are showing the same information of the biases (underestimation or overestimation) of the simulations in relation to the observed precipitation. In addition, Figure 4 has different scales in each season and it is not easy to compare them. In this way, it should be better to combine the top panels of Figure 4 with the relative biases in Figure 5 to have only one Figure showing the biases.

Done, Thanks. They have been combined in Figure 3, and the discussion has been updated in L218-224

- 5. The statements in L19-20 do not correspond to what is presented in the manuscript. In the actual form, no sensitivity numerical tests (by changing the core from hydrostatic to non-hydrostatic or increasing/reducing the number of vertical levels) or the circulation pattern were presented in the manuscript. I am suggesting: 1) to remove "Sensitivity analysis ... precipitation outputs"; 2) to include in the manuscript an analysis of the circulation patterns (point 2) to justify the affirmation in the end of phrase.
- 1) Done, Thanks.
- 2) Done in L19, Thanks.
- 6. To improve understanding, the authors could reorganize the introduction. Suggestion:

Starting with L24-27 and following with L37-45. After that, L27-37 would be followed by L46-58.

Done. Thanks.

7. I noted that the main objective of the text is not clearly established at the end of the introduction. There are some indications of the objectives in L64-67. I suggest that the authors reorganize the objectives clearly at the end of introduction.

We appreciate the reviewer's helpful observation. The end of the introduction has been revised to explicitly present the main objective and motivation of the study in L92-97.

**Minor points**

In many parts of the text (L31; L46; L142-150; L195; L265), the citations are not correct. For example: L108 - in (Giorgi et al, 2023b) should be in Giorgi et al. (2023b); L148 should be "by Holslag et al. (1990)"; L149 should be "by Zeng et al. (1998)" and many others in the text. Please, check all the text.

We thank the referee for the careful reading and for pointing this out. We would like to clarify that the in-text citations in our manuscript follow the Copernicus Publications reference style. For example, the journal specifies "(Smith et al., 2021)" rather than "Smith et al. (2021)".

L30-31 - should be "... interannual) controlled by the Global .... Abtew et al. (2019) found ... "

Done in L39, Thanks.

L37 - should be " ... rainfall since it is the main rain ... "L39;

I think this was in L36. I want to clarify that "... as a main rain producing system ..." belongs to the ITCZ. It appears that it is not clear, so the sentence is paraphrased in L47.

L109-111 - should be "are driven by atmospheric variables and SST from ERA5 reanalysis data from ECMWF (Hersbach et al., 2020) with 0.250 x 0.250 of horizontal resolution for the period 2000-2009. For evaluation, observed ... for the period 2001-2009" since SST was already defined in L31.

Done in L109. Thanks,

L118 - should be "The domain has a 10 km horizontal ... longitude (Fig.2), involving ...""

Done in L18, Thanks,

**L121 - The 18 vertical levels is a very small number for a 10km horizontal resolution. Why only 18 vertical levels?**

We thank the reviewer for this important point. RegCMs have a long history of applications using 18 sigma vertical levels as the default configuration and with a model top of 50 hPa. This configuration has been used across a range of horizontal resolutions, including high-resolution of 10 km, because it represents a compromise between vertical resolution and the substantial computational cost of multi-year/high-resolution runs. For example, previous RegCMs applications have used 18 sigma levels at 10 km and comparable resolutions (e.g., (Torma et al., 2011)). We fully agree that for very high-resolution or convection-permitting simulations (grid spacing less than 10 km up to  $\leq$  4 km), it is recommended and necessary for accuracy. This is because the sigma-coordinate system compresses in meters as moving up, meaning more levels are needed to provide sufficient vertical resolution near the surface. However, for the current hydrostatic 10 km configuration, the 18-level setup remains within the standard and widely validated range used in many RegCM studies.

L127 - should be "... in the different numerical experiments ..." since the authors used the code as it is, i.e., they do not change any parameter or physical parametrization.

Done in L129-130. Thanks.

L139-140 - should be "Hence, a new set of simulations was conducted by using Nogherotto-Tompkins (NoTo; Nogherotto et al., 2016) microphysic scheme, which treats the mixed .... Over East Africa, Godoshava and Semazzi (2019) revealed .... In addition, Kalmár et al. (2021) ..."

Done in L140-141. Thanks.

L150 - In many parts of the text, the authors used Emanuel\_NoTo, Tiedke\_Noto and Kain-Fritsch\_NoTo. This information should be in Table 2.

We thank the referee for the suggestion. As recommended, we have revised Table 2 lin L155 and added a new column specifying the scenario names.

L157 - remove "calibration"; it is more common to use "relative bias" instead "percent bias".

Done in L158. Thanks.

L158 - What is the interpretation of the RSR index? What is the range of acceptance of RSR? It was calculated considering a time series or for the mean spatial pattern in UBNB? Please, clarify it in the text and also include a reference for the index.

We thank the reviewer for this valuable comment. The RSR (RMSE–observations standard deviation ratio) index was calculated following (Moriasi et al., 2007). The RSR is defined as the ratio of the root mean square error (RMSE) to the standard deviation of the observed data, and it provides a standardized measure of model performance where lower values indicate better agreement between simulated and observed values. The RSR values range from 0 to  $\infty$ , with 0 indicating a perfect match between simulation and observation. According to (Moriasi et al., 2007), model performance is considered satisfactory when RSR  $\leq$  0.70.

In our study, we first computed the monthly mean spatial precipitation over the UBNB, then used this basin-averaged monthly time series to calculate the RSR between simulated and observed precipitation. The requested clarification and the reference have been added to the revised manuscript in L160-162.

L163 - remove "estimated"

If the word is deleted, the meaning of the sentence is not complete.

L165 - remove "to check spatiotemporal distribution" since this information is already in the beginning of the phrase.

Done. Thanks.

L180 - should be " ... for the period 2001-2009 since the year 2000 was considered as ..."

Done in L186. Thanks.

L190 - What is the meaning of "reduces the significance between CCs"? Please, clarify in the text.

It is modified to be clear in L196.

L194 - should be " are presented in Fig. 3b".

Done in L200. Thanks.

L183 - should be " ... UBNB the Fig. 3 and Table 3 show the evaluation of the simulated precipitation for the seven numerical experiments""

Done in L190. Thanks.

L202 - Suggestion "The performance of the experiments S5, S6 and S7, which use the NoTo microphysics scheme, is evaluated by analyzing the spatial pattern ..."

Thanks a lot for the suggestion. It is done in L216.

L206 - should be "The model also overestimated rainfall in eastern ..."

Done in L218. Thanks.

L218 - should be "need an improvement"

Done in L234. Thanks.

L221 - should be "exhibit a low correlation"

Done in L238. Thanks.

L242-243 - should be "characteristics simulated in the experiments S5, S6 and S7 and observed by PP7 during the FMAM, JJAS and ONDJ (Fig. 7). "

Done in L244. Thanks.

L245 - should be "... (S6) slightly better represents the observed distribution of the daily precipitation during the FMAM (Fig. 7a)"

Done in L246. Thanks.

L278 - I think that should be better "non-convective precipitation and reduces the biases in the simulated total precipitation."

Done in L331. Thanks.

L296 - What is the meaning of "refine the physical parametrizations"?

We mean to investigate and test more physical parametrizations that affect the precipitation simulation. It is modified in the revised manuscript in L353. Thanks.

**Figures:**

1) The labels in most of the figures need to be improved since they are very small, which makes it difficult for readers to interpret the figures.

The figures have been improved. Thanks.

2) Please, highlight the UBNB basin (shown in Fig. 1) in panel (a) of Figure 2.

Done in L124. Thanks.

**References**

Giorgi, F., Coppola, E., Giuliani, G., Ciarlo`, J. M., Pichelli, E., Nogherotto, R., Raffaele, F., Malguzzi, P., Davolio, S., Stocchi, P., and Drofa, O.: The Fifth Generation Regional Climate Modeling System, RegCM5: Description and Illustrative Examples at Parameterized Convection and Convection-Permitting Resolutions, Journal of Geophysical Research: Atmospheres, 128, e2022JD038199, 2023.

Moriasi, D. N., Arnold, J. G., Liew, M. W. Van, Bingner, R. L., Harmel, R. D., and Veith, T. L.: Model Evaluation Guidlines for Systematic Quantification of Accuracy in Watershed Simulations, Trans ASABE, 50, 885–900, 2007.

Osman, M., Zittis, G., Haggag, M., Abdeldayem, A. W., and Lelieveld, J.: Optimizing Regional Climate Model Output for Hydro-Climate Applications in the Eastern Nile Basin, Earth Systems and Environment, 5, 185–200, https://doi.org/10.1007/S41748-021-00222-9/FIGURES/11, 2021.

Prein, A. F., Gobiet, A., Truhetz, H., Keuler, K., Goergen, K., Teichmann, C., Fox Maule, C., van Meijgaard, E., Déqué, M., Nikulin, G., Vautard, R., Colette, A., Kjellström, E., and Jacob, D.: Precipitation in the EURO-CORDEX 0.11° and 0.44° simulations: high resolution, high benefits?, Clim Dyn, 46, 383–412, https://doi.org/10.1007/S00382-015-2589-Y/FIGURES/19, 2016.

Torma, C., Coppola, E., Giorgi, F., Bartholy, J., and Pongrácz, R.: Validation of a High-Resolution Version of the Regional Climate Model RegCM3 over the Carpathian Basin, J Hydrometeorol, 12, 84–100, https://doi.org/10.1175/2010JHM1234.1, 2011.

Torma, C., Giorgi, F., and Coppola, E.: Added value of regional climate modeling over areas characterized by complex terrain-precipitation over the Alps, J Geophys Res, 120, 3957–3972, https://doi.org/10.1002/2014JD022781;REQUESTEDJOURNAL:JOURNAL:21698996;WEBSITE:WEBSITE:A GUPUBS;JOURNAL:JOURNAL:21562202D;WGROUP:STRING:PUBLICATION, 2015.

---

## Author Comment (AC2)

**Reply on Referee #2 Comments**

This manuscript investigates the performance of the RegCM5 model with a hydrostatic dynamical core in simulating precipitation over the complex terrain of the Upper Blue Nile Basin (UBNB). The study leverages high-resolution ERA5 reanalysis data to evaluate various convective and microphysical parameterization schemes, ultimately identifying the Emanuel convection scheme coupled with the Nogherotto-Tompkins (NoTo) microphysics scheme as the optimal configuration. The following points should be addressed to improve the clarity, robustness, and overall quality of the manuscript:

**Major points:**

1. The statement that even the best parameterization shows a 66% bias is significant. This level of error raises concerns about the model's credibility for the region. The authors should provide a more comprehensive discussion on how to interpret this result. Specifically, they should: a) Clearly state what level of bias is considered acceptable for the intended applications of the model in this study. b) Discuss whether the model, despite the bias, successfully captures the dominant spatial and temporal patterns of climate variability, which can sometimes be more important than the absolute values for certain applications.

We thank the reviewer for this valuable comment. We have revised the discussion part in the results and discussion section to provide a clearer interpretation of the model bias and to explain its implications for the intended applications of this study. Specifically, we clarified that there is no universally accepted threshold for "acceptable" bias in RCM simulations. Instead, many previous studies have emphasized that the model's ability to reproduce the dominant spatial and temporal variability of climate variables is often more critical than the absolute magnitude of bias (L203-213). In addition, based on that, the abstract was also modified in L16-17.

2. The methodology does not mention a threshold for defining a "wet day" (e.g., >0.1 mm/day). Please clarify if all precipitation values, including trace amounts, were used in the analysis. The low frequency of zero precipitation values shown in Figure 7 suggests the model (and possibly the observations) rarely simulates completely dry days. Confirming the wet-day threshold is important for interpreting intensity and frequency results.

We thank the reviewer for this important observation. In the initial analysis, all precipitation values, including trace amounts, were considered. Following the reviewer's recommendation, we have revised the methodology and identified a wet-day threshold of precipitation > 1 mm/day to exclude trace rainfall and ensure consistency in interpreting intensity and frequency analyses. This clarification has been added to the methodology section (L175-177). Consequently, the figure (L247) has been updated based on this threshold. The overall interpretation of the results remains essentially unchanged and consistent with the previous version.

3. The manuscript compares different parameterizations but lacks a detailed discussion on the physical reasons why the optimal scheme (Emanuel + NoTo) performed best. The authors should elaborate on the key physical differences between the schemes (e.g., in their triggering of convection, closure assumptions, or ice-phase processes) and provide a hypothesis for why this specific combination is more suitable for representing the dominant atmospheric processes in the UBNB.

We sincerely thank the reviewer for this valuable suggestion. Accordingly, we have added a detailed explanation of the physical reasons behind the superior performance of the Emanuel + NoTo combination in the discussion part in the results and discussion section (L306-313). Specifically, we discuss how the Emanuel convective scheme, which represents convection as a buoyancy-driven, quasi-equilibrium process with entrainment—detrainment effects and moisture adjustment, tends to better capture the deep convective systems prevalent over UBNB. When coupled with the NoTo large-scale condensation scheme, known for its improved representation of microphysical processes and reduced excessive condensation under weakly saturated conditions, the combination provides a more realistic simulation of both convective and stratiform rainfall.

**Minor points:**

1. The 10km resolution may be insufficient to capture local processes in the UBNB's complex terrain. Justification for this choice (e.g., computational limits or sensitivity tests) should be provided, along with discussion on how resolution impacts simulation accuracy.

The chosen resolution of 10 km represents a common mesoscale configuration in regional climate modeling that balances physical realism and computational feasibility. This resolution has been shown to enhance the representation of precipitation, especially over complex terrain. In earlier studies that relied on ERA-Interim reanalysis, achieving such resolutions typically required double nesting (e.g., downscaling from ~50 km to ~12–10 km) (e.g., (Prein et al., 2016; Torma et al., 2015)). In addition, the tropical climate of the UBNB, where rainfall is dominated by seasonal systems, a 10 km resolution provides sufficient fidelity for the current study. Nonetheless, for some applications, such as assessing land use and land cover change impacts (e.g., due to new reservoirs impounded upstream, existing and potential dams along the basin), there is a need for a higher resolution. Maybe in future work, we will exploit the MOLOCH non-hydrostatic core and test it for using the convective permitting approach that will enable us to better represent the very small change in land surface features that couldn't be represented in lower resolutions and thereby capture any change in the localized land-atmosphere interactions.

2. Figures 1 and 2 should be merged to clearly show the UBNB within the larger model domain, improving readability.

**Done in L124, Thanks.**

3. L158 - Although monthly evaluation is common, adding daily-scale metrics would strengthen the assessment of rainfall distribution and intensity.

We appreciate the reviewer's valuable suggestion. In fact, daily-scale analysis was also conducted in our study. The probability density function (PDF) analysis, which evaluates the distribution and intensity of daily precipitation, was performed based on daily data and is already described in the Methodology section. Accordingly, we clarified this point in the manuscript to make it more explicit.

4. The spatial discontinuities in simulated precipitation (Fig. 4), especially in the northeast, require physical explanation—g., topography representation, land-atmosphere feedbacks, or convective processes.

We thank the reviewer for this insightful comment. Following Reviewer 1's suggestion, the spatial distribution figure (previously Figure 4 and now Figure 3) was removed and combined with the bias plots to avoid redundancy, as both presented similar information. However, the spatial discontinuities mentioned by the reviewer remain visible in the updated bias plots, which helped us to further interpret their causes. The discontinuities are mainly attributed to the strong topographic gradients and land-atmosphere interactions over the northeastern highlands of the basin, where local convective triggering is highly sensitive to terrain-induced uplift and land surface heterogeneity. This is added in L221-224.

5. The selection of Emanuel+NoTo needs clarification, as Kain-Fritsch+NoTo performed well in explained variance for key seasons (Fig. 6). The trade-offs and reasons for the final choice should be explicitly explained.

We thank the reviewer for this valuable comment. While the Kain–Fritsch + NoTo configuration indeed showed a relatively higher explained variance during the short rain (FMAM) and dry (ONDJ) seasons, it performed poorly during the long rain season (JJAS), which contributes approximately 70% of the total annual precipitation over the UBNB. However, the text has been modified in L235.

**References**

Prein, A. F., Gobiet, A., Truhetz, H., Keuler, K., Goergen, K., Teichmann, C., Fox Maule, C., van Meijgaard, E., Déqué, M., Nikulin, G., Vautard, R., Colette, A., Kjellström, E., and Jacob, D.: Precipitation in the EURO-CORDEX 0.11° and 0.44° simulations: high resolution, high benefits?, Clim Dyn, 46, 383–412, https://doi.org/10.1007/S00382-015-2589-Y/FIGURES/19, 2016.

Torma, C., Giorgi, F., and Coppola, E.: Added value of regional climate modeling over areas characterized by complex terrain-precipitation over the Alps, J Geophys Res, 120, 3957–3972, https://doi.org/10.1002/2014JD022781;REQUESTEDJOURNAL:JOURNAL:21698996;WEBSITE:WEBSITE:A GUPUBS;JOURNAL:JOURNAL:21562202D;WGROUP:STRING:PUBLICATION, 2015.